# Effects of Slag Composition and Impurities of Alloys on the Inclusion Transformation during Industrial Ladle Furnace Refining

**Chunyang Liu [1], Yi Jia [1], Lixia Hao [1], Shaowei Han [1], Fuxiang Huang [1], Huixiang Yu [2], Xu Gao [3,*,†], Shigeru Ueda [4] and Shin-ya Kitamura [5,†]**

[1] Steelmaking Plant of Beijing Shougang Co., Ltd., No. 25 Zhao'an Street, Western Industrial Zone, Qian'an, Tangshan 064400, China; chunyangliuvip@163.com (C.L.); jiayi@sgqg.com (Y.J.); haolixia@sgqg.com (L.H.); hanshaowei2016@163.com (S.H.); huangfx3232@sgqg.com (F.H.)

[2] School of Metallurgical and Ecological Engineering, University of Science & Technology Beijing, 30 Xueyuan Road, Haidian Distinct, Beijing 100083, China; yuhuixiang@ustb.edu.cn

[3] School of Metallurgy and Environment, Central South University, Changsha 410083, China

[4] Institute of Multidisciplinary Research on Advanced Materials, Tohoku University, 2-1-1 Katahira, Aoba ku, Sendai 980-8577, Japan; tie@tohoku.ac.jp

[5] Emeritus Professor of Tohoku University, Sendai 980-8577, Japan; shinya.kitamura.e7@tohoku.ac.jp

\* Correspondence: xgao.sme@csu.edu.cn

† Previously at the Institute of Multidisciplinary Research on Advanced Materials, Tohoku University, 2-1-1 Katahira, Aoba ku, Sendai 980–8577, Japan.

**Abstract:** The inclusion of the $MgO \cdot Al_2O_3$ (MA) spinel and $CaO–Al_2O_3$ are occasionally observed during the refining of Al–killed steel, even without the intentional additions of Ca and Mg. Many studies have focused on the source of Mg and Ca; however, especially for the formation of $CaO–Al_2O_3$–type inclusions, some recent results showed that Ca was difficult to dissolve from refining slag, even when the Al content in molten steel was high. To confirm these differences, industrial experiments were designed in this study, and the effects of the FeO and MnO contents, as well as the impurities of the alloying materials, were discussed. The results showed that, when the FeO and MnO contents in slag were high (about 10 mass%) at the start of LF (ladle furnace), all inclusions remained as $Al_2O_3$, despite alloying. Using the slag with low FeO + MnO content (<1~2 mass%), the $Al_2O_3$ inclusions changed to the MA spinel, but $CaO–Al_2O_3$ inclusions were not observed, indicating that $CaO–Al_2O_3$ inclusions were difficult to form by the steel/slag reactions under the current conditions. Only for the molten steel that contained a low level of dissolved oxygen and a large amount of Fe–Si, which contained Ca as the impurity was added, $CaO–Al_2O_3$ inclusions were generated.

**Keywords:** inclusion transformation; $Al_2O_3$; spinel; $CaO–Al_2O_3$

## 1. Introduction

In the refining process of clean steel, $MgO–Al_2O_3$–type (MA spinel) and $CaO–Al_2O_3$–type inclusions sometimes formed after Al deoxidation, even without the intentional addition of Mg or Ca to molten steel. Many researchers have reported this phenomenon [1,2], and the source of both Mg and Ca to transform the $Al_2O_3$ inclusion have been studied. About the transformation of $Al_2O_3$ to MA spinel inclusion, the Mg source has been considered as both the refining slag and the refractories. In the case of refining slag, the slag composition affects much in the formation of a MA spinel. By reacting 15–20 kg of molten stainless steel with refining slag, T. Nishi et al. [3] and G. Okuyama et al. [4] clarified that the $CaO/SiO_2$ and $CaO/Al_2O_3$ ratios of slag increased the MgO content in the inclusions. M. Jiang et al. [5] studied the effects of $Al_2O_3$ and MgO activities on the inclusion transformation in a high–carbon alloyed steel. A. Harada et al. [6–8] optimized the $CaO/SiO_2$ and $CaO/Al_2O_3$ ratios and established a kinetic model to precisely estimate the inclusion

transformation. In the case of refractory, the dissolution of Mg from various refractory materials has been studied, such as MgO–C [9–12], MgO [13], magnesia–chromite [14], and dolomite [15]. Since the dissolution rate of Mg from refining slag was higher than the refractory [15], the major Mg source should be the refining slag. However, the influence of the refractory must not be ignored. For the formation of CaO–Al$_2$O$_3$–type inclusion, some studies using laboratory–scale experiment proposed that CaO in the top refining slag was reduced by Al dissolved in motel steel and the release of Ca to molten steel to transform the inclusions [16,17]. In some industrial tests, CaO–Al$_2$O$_3$–type inclusions were observed at the end of the LF (ladle furnace) treatment when the Al content in molten steel was lower than 0.25mass% [18–20], and using a thermodynamic estimation, these studies deduced that the source of Ca to transform inclusions was from the reduction of slag. On the other hand, some researchers proposed that the origin of the CaO–Al$_2$O$_3$–type inclusion was the entrapped droplets of refining slag [13]. According to a previous study of some of the authors [21], only 0.3 ppm of Ca was reduced from a CaO saturated slag by a molten steel containing 0.25 mass% of Al, and CaO–Al$_2$O$_3$–type inclusions were not observed; when the Al content in steel increased to about 0.75 mass%, the Ca content in molten steel increased to 0.9 ppm and CaO–Al$_2$O$_3$–type inclusion formed slowly after 120 min of reaction. This result indicated that, for a steel with normal grade of Al content, the CaO–Al$_2$O$_3$–type inclusion is difficult to form by the slag/steel reaction. In addition, using equilibrium and kinetic discussions, Kumar et al. [22] also found that Ca dissolution from refining slag by Al reduction was small and affected little on the inclusion transformation. Therefore, different opinions on the Ca source to generate CaO–Al$_2$O$_3$–type inclusions exist. Since most of the studies were limited to a laboratory scale, and the thermodynamic date, especially for the Ca–O equilibrium, varied largely, owing to the measuring conditions [2], an industrial test was necessary to confirm this phenomenon.

To design the industrial test, we focused on the effects of refining slag composition and another possible Ca source—the impurities of alloying materials. The refining slag composition determines both the oxygen potential of the slag/metal interface and the inclusion adsorbing ability; thus, many studies have been conducted. Recently, A. Harada et.al. [7,8] studied the effects of CaO/Al$_2$O$_3$ and CaO/SiO$_2$ ratios on the transformation of Al$_2$O$_3$ to MA spinel inclusion. They found a decrease in MgO activity of the slag by lowering the CaO/Al$_2$O$_3$ and CaO/SiO$_2$ ratios, which could suppress the formation of MA spinel inclusion. J. H. Shin et al. [23] reported the effects of the CaO/Al$_2$O$_3$ ratio of the refining slag on the transforming behavior of inclusions in a Mn–V–alloyed steel. They concluded when the CaO/Al$_2$O$_3$ ratio was between 1.5 to 2.5, the transfer of Mg and Ca from slag to molten steel generated liquid inclusions. As the CaO/Al$_2$O$_3$ ratios have been widely investigated, we focused on the FeO and MnO content, because it affects the oxygen potential of the slag/metal interface. Several studies clarified FeO in refining slag, which was carried over from converter slag and influenced the cleanness of molten steel and the defects of rolled sheet [24,25]. However, there are few studies focusing on the transforming behaviors of Al$_2$O$_3$ inclusion by the change in the FeO and MnO contents of slag.

As another possible source of Ca, the alloy materials have also been studied. Both K. Mizuno et al. [26] and O. Wijk et al. [27] confirmed the Al and Ca in ferrosilicon affected the transformation of inclusions by laboratory–scale experiments. In practical refining, many other alloy materials are added together with ferrosilicon, and the timing of Al deoxidation can be altered even after ferrosilicon addition. Under such circumstances, it is necessary to confirm the effects of the impurities in alloying materials using industrial tests.

Therefore, in this study, industrial experiments were conducted in 210–300 tons of ladle furnace refining. In these experiments, the change in inclusion transforming behavior by the FeO and MnO contents of refining slag, and by the impurities of the alloying materials, were studied.

## 2. Experimental Method

### 2.1. Experimental Apparatus and Procedure

In this study, 4 heats of the experiment were conducted during LF refining, and the experimental set-up is shown in Figure 1. The capacity of the ladle is 210 tons for heats 1, 2 and 4 and 300 tons for heat 3. All the experiments were conducted with graphite electrode heating and the strong intensity of bottom stirring. The lining material of the ladle, contacted with steel, was C containing $Al_2O_3$–MgO–type brick, and its composition was 35 mass%$Al_2O_3$–50 mass%MgO–10 mass% C.

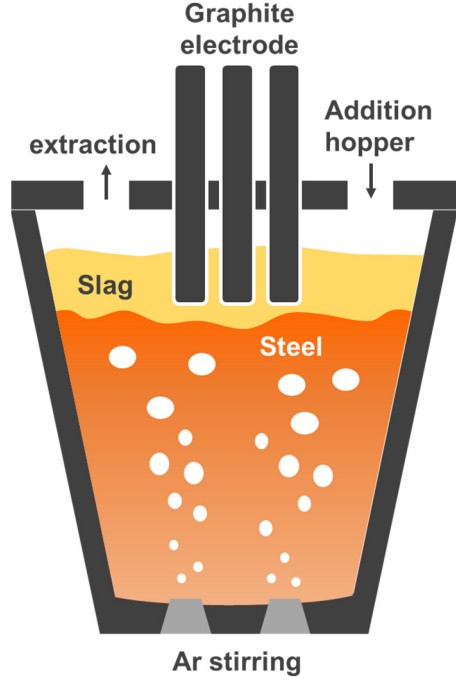

**Figure 1.** Schematic diagram of the LF (ladle furnace) experiments.

The production practice of the LF treatment together with the sampling method is shown in Figure 2. After BOF (Basic Oxygen Furnace) refining, the alloy; Al deoxidizer; and slag former (oxides like lime, alumina, magnesia, etc.) were added into molten steel during tapping, and then, the LF treatment began. The main components of slag were CaO, $SiO_2$, $Al_2O_3$, MgO, FeO, and MnO. It had to be noticed that the oxides like FeO and MnO were not intentionally added but were carried over from converter slag during tapping. During the LF treatment, alloy adjustments were conducted twice, which are labeled as the 1st and 2nd alloys in Figure 2. In each treatment, 6 steel samples were taken, which are labeled as S1–S6 in Figure 2. S2 and S4 were taken at 3 min before adding the alloys, and S3 and S5 were taken at 5 min after alloying. At the same time, the refining slag was also sampled. Steel samples were taken using a previously designed mold and following same procedure [28], and the slag samples were taken by immersing a cold steel rod. The dissolved oxygen content and temperature in steel were measured at the start of treatment and after the 1st and 2nd alloying. The dissolved oxygen content was measured by a solid electrode sensor with $Cr/Cr_2O_3$ reference. In these experiments, the steel temperature was stabilized at 1848–1873 K. The operating conditions of the 4 heats are listed in Table 1, and the alloying conditions are summarized in Table 2.

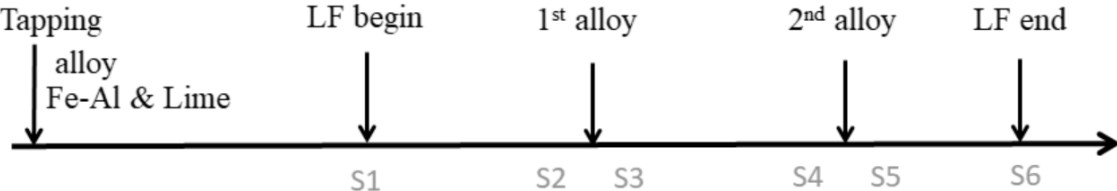

PS: S2, S4: just before alloy; S3, S5: 3–5 min after alloy;

**Figure 2.** LF production practice and the sampling plan.

**Table 1.** Experimental conditions.

| Experiment NO. | Slag Condition | Alloy Amount | Alloy Time/Min | Deoxidizer Added during Alloying |
|---|---|---|---|---|
| Heat 1 | High FeO content at the start of treatment; MgO & CaO sat. | Small | 1st alloy: 20 | Al |
| Heat 2 | Low FeO content at the start of treatment; MgO & CaO sat. | Small | 1st alloy: 20<br>2nd alloy: 40 | Al only<br>No deoxidizer |
| Heat 3 | Low FeO content at the start of treatment; MgO & CaO sat. | Large | 1st alloy: 13<br>2nd alloy: 28 | Al<br>Fe–Si |
| Heat 4 | Low FeO content at the start of treatment; MgO & CaO sat. | Large | 1st alloy: 30<br>2nd alloy: 60 | Fe–Si<br>Fe–Al |

**Table 2.** Amount and timing of the added alloys in each heat.

| Experiment No. | During BOF Tapping kg/ton | During 1st Alloy kg/ton | During 2nd Alloy kg/ton |
|---|---|---|---|
| Heat 1 | Fe–Al 3.37<br>Fe–Mn 1.64 | Fe–Al 0.85<br>C 0.10 | – |
| Heat 2 | Fe–Al 4.92 | Fe–Al 0.91 | Fe–Mn 1.20 |
| Heat 3 | C 0.17<br>Fe–Mn 3.92<br>Fe–Si 3.29<br>Fe–P 1.67<br>Al 1.97<br>Fe–Cr 5.93<br>Ni 0.66<br>Cu 2.79 | C 0.08<br>Al 0.53<br>Fe–Si 0.66<br>Fe–Cr 1.48<br>Fe–P 0.64 | Fe–P 0.28<br>Fe–Mn 0.23<br>Fe–Si 0.43<br>Al 0.20 |
| Heat 4 | C 6.41<br>Fe–Al 3.43<br>Fe–Si 4.21<br>Fe–Mn 13.16 | Fe–Mn 2.02<br>Fe–Si 0.25<br>C 0.38 | Fe–Mn 0.48<br>Fe–Al 0.35<br>C 0.23 |

*2.2. Analysis Method*

The obtained steel samples were quenched by water and cut for both the chemical composition analysis and inclusion observation. The cutting method of the steel sample is shown in Figure 3.

For the chemical analysis, the Al and Mg contents of the steel sample were analyzed using inductively coupled plasma atomic emission spectroscopy (ICP-AES, SPECTRO); the Ca content of the sample was analyzed by glow discharge mass spectrometry (GDMS, Thermo Scientific). The other alloy elements in steel were measured by the spark discharge emission spectrometry method. The total oxygen (T.O) of the steel sample was measured using the infrared X-ray absorption method.

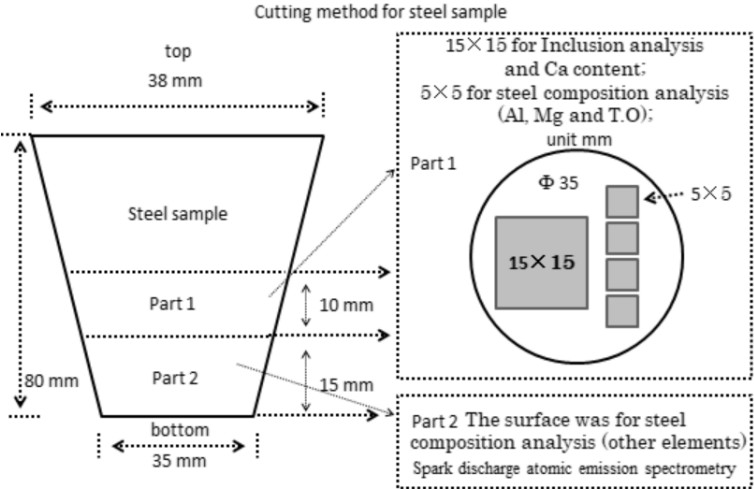

**Figure 3.** Cutting method of the steel sample.

For the inclusion observation, a scanning electron microscope (SEM) equipped with an automatic inclusion analysis system was used (P-SEM, ASPEX corporation). During the P-SEM analysis, Mg, Al, Fe, Ca, O, S, and Si were selected as target elements, since they existed in the steel sample. The inclusions were assumed to consist of MgO, $Al_2O_3$, and CaO, and the analyzed values of Mg, Al, and Ca were converted to oxide values using a stoichiometric relationship. The value of Fe was ignored, because it possibly came from the metal phase rather than inclusion. The compositions of the sampled slags were measured by the X-ray fluorescence spectrometer (XRF, RIGAKU) method.

All the above analysis except the P-SEM were conducted at the Institute of Multidisciplinary Research for Advanced Materials (IMRAM), Tohoku University, Sendai, Japan. The P-SEM analysis was conduct at the University of Science and Technology Beijing (USTB).

*2.3. Alloying Materials*

The compositions of all the alloying materials used in this study are shown in Table 3. To study the impurities of the alloying materials on the inclusion transformation, both the Mg and Ca contents of each alloy were analyzed using ICP-AES. The microstructure of the alloy was observed by Field Emission-Electron Probe Micro-Analyzer (FE-EPMA, HITACHI, Sourced by Tohoku University, Sendai, Japan).

**Table 3.** Alloying materials and the Mg and Ca impurities.

| Alloy Brand | Mg Content/Mass% | Ca Content/Mass% |
|---|---|---|
| Fe–Al | 0.0003 | 0.0010 |
| Fe–Mn–high C | 0 | 0.0010 |
| Al (lump) | 0.0628 | 0.1910 |
| Fe–Mn–medium C | 0.0003 | 0.0018 |
| Fe–Cr | 0 | 0 |
| Fe–P | 0 | 0.0008 |
| Fe–Si | 0.0261 | 0.3195 |

As shown in Table 3, only Al (lump) and Fe–Si contain the impurities of Mg and Ca. The microstructures of Al (lump) and Fe–Si are shown in Figures 4 and 5, respectively. For Al (lump), both Ca and Mg mainly exist in the oxide phases, which are bright phases and dark phases in Figure 4. For the Fe–Si as shown in Figure 5, Mg coexists with Ca in a phase that mainly contains Si and Ca. In addition, the Fe–Si phase also contains certain amounts of Ca.

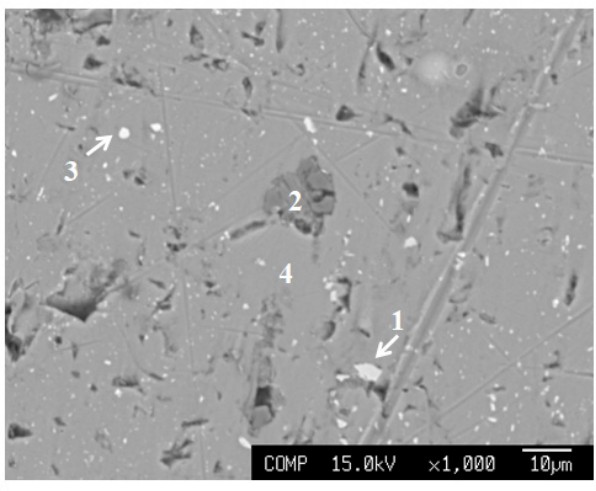

| Position | Content / mass% | | | | | |
|----------|------|-------|-------|-------|-------|-------|
|          | Mg   | Al    | Si    | Ca    | Fe    | O     |
| 1#       | 0.67 | 4.33  | 15.49 | 10.28 | 28.63 | 31.85 |
| 2#       | 1.08 | 22.28 | 17.41 | 0.25  | 4.94  | 41.72 |
| 3#       | 0    | 67.33 | 30.38 | 0     | 2.29  | 0     |
| 4#       | 0    | 99.71 | 0.11  | 0     | 0.18  | 0     |

**Figure 4.** Microstructure of Al (lump).

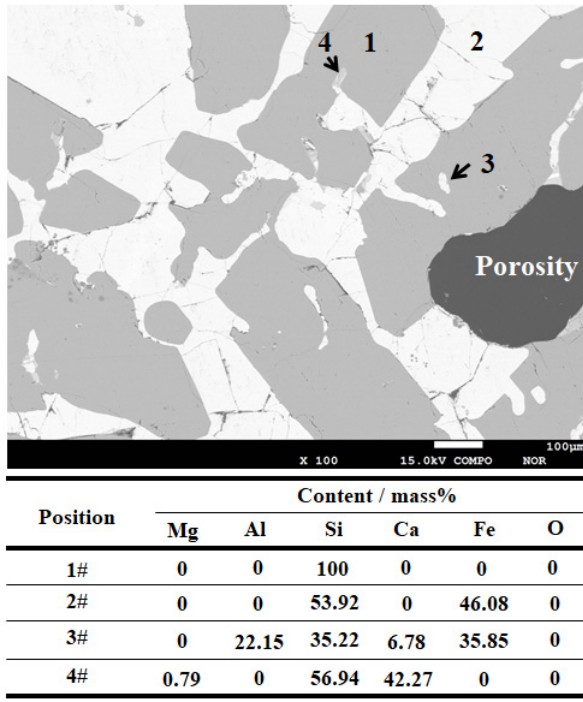

| Position | Content / mass% | | | | | |
|----------|------|-------|-------|-------|-------|---|
|          | Mg   | Al    | Si    | Ca    | Fe    | O |
| 1#       | 0    | 0     | 100   | 0     | 0     | 0 |
| 2#       | 0    | 0     | 53.92 | 0     | 46.08 | 0 |
| 3#       | 0    | 22.15 | 35.22 | 6.78  | 35.85 | 0 |
| 4#       | 0.79 | 0     | 56.94 | 42.27 | 0     | 0 |

**Figure 5.** Microstructure of the Fe–Si alloy.

## 3. Experimental Results

### 3.1. Steel Composition

The differences in Al, as well as the oxygen contents between heat 1 and heat 2 are shown in Figure 6. These two heats are compared here because similar procedures of deoxidation were conducted, and the main difference was the FeO and MnO contents of the top slag. Throughout the LF treatment, the Al contents of heat 2 were higher than heat 1. This was because the added amount of the Fe–Al alloy during BOF tapping and

1st alloying for heat 2 was larger than heat 1, and the reduction of FeO and MnO in slag consumed Al in the case of heat 1. The total oxygen contents of heat 1 were slightly higher than heat 2, and both decreased with time because of the inclusion removal. Besides, the content of dissolved oxygen of heat 1 was about 7.5 ppm, while, for heat 2, it was about 5.5 ppm. Therefore, a more sufficient deoxidation was accomplished for heat 2 compared to heat 1, and the content of the dissolved Al in molten steel in the case of heat 1 was much lower than heat 2.

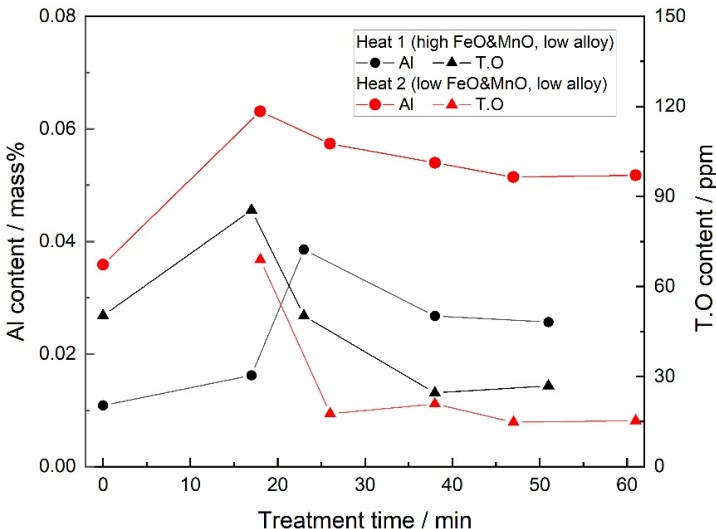

**Figure 6.** Comparison on the Al and total oxygen contents between heat 1 and heat 2.

The Ca and Mg contents of heat1 and heat 2 are compared in Figure 7. Owing to the differences in the Al contents, the Mg content of heat 2 was higher than heat 1. For both heats, Mg was not intentionally added or through alloy addition; thus, Mg was reduced from the top slag or refractory by the dissolved Al in molten steel, and a higher Al content led to a higher Mg content. However, the Ca contents were low and similar for both heats, despite the differences in the Al contents. This indicated that the reduction rate of CaO from the top slag by dissolved Al in molten steel was extremely slow.

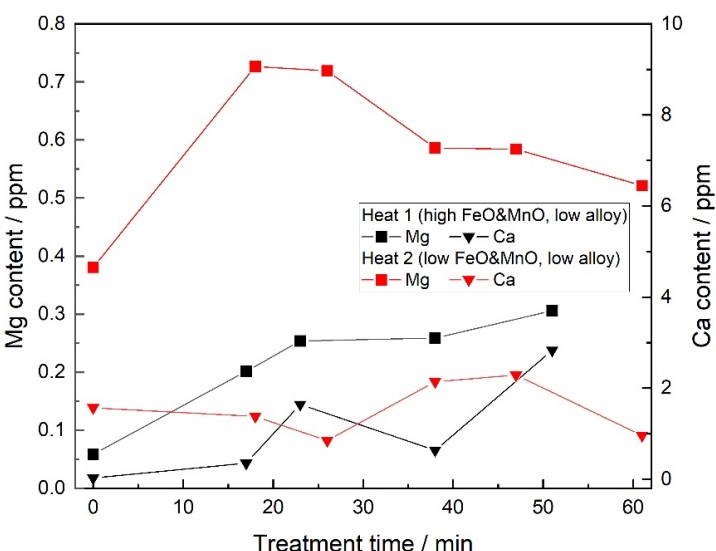

**Figure 7.** Comparison on the Ca and Mg contents between heat 1 and heat 2.

For the heats with large additions of allying materials, the Al, as well as total oxygen contents, for heat 3 and heat 4 are compared in Figure 8. For heat 3, after the 1st alloying,

which was conducted at 13 min, the Al content increased and reached a similar level with heat 4. With a further addition of Al during the 2nd alloy addition near the LF end for heat 3, the Al content changed little and was kept at about 0.04 mass%. The total oxygen content of heat 3 was higher than heat 4. For heat 4, because none of the Al–containing materials was added during the 1st alloying, and the 2nd alloying was near the LF end, the Al content changed little. The total oxygen decreased from 26 ppm to 15 ppm, owing to the removal of the inclusions. Compared to Figure 6, the Al contents of heat 3 and heat 4 were higher than heat 1 and slightly lower than heat 2.

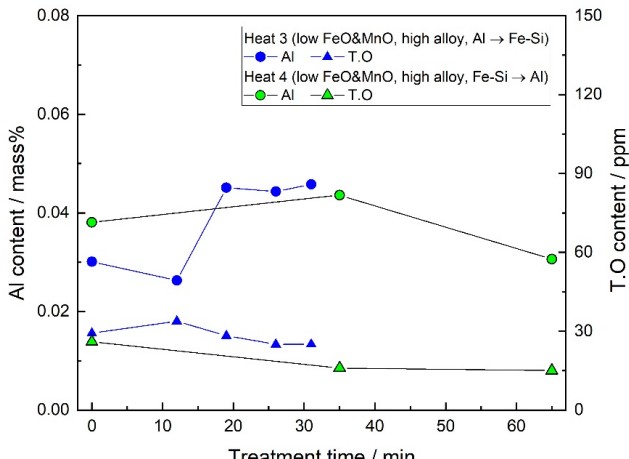

**Figure 8.** Comparison of the Al and total oxygen contents between heat 3 and heat 4.

The Mg and Ca contents of heat 3 and heat 4 are compared in Figure 9. Compared to heat 1 and heat 2, both the Mg and Ca contents of heat 3 were much higher. As the Al content of all heats were at the same order of magnitude, the reduction of slag by dissolved Al would not lead to such a large increase in the Mg and Ca contents. As mentioned in Table 3 and Figure 5, the Fe–Si alloy contained metallic Mg and Ca as impurities, and it was added to heat 3 from BOF tapping to the 2nd alloying. Therefore, the main source of Mg and Ca for heat 3 was the Fe–Si alloy. For heat 4, the Ca content was lower than heat 3 and similar to heat 1 and heat 2 because of the smaller addition of the Fe–Si alloy for heat 4, especially during the 1st alloying.

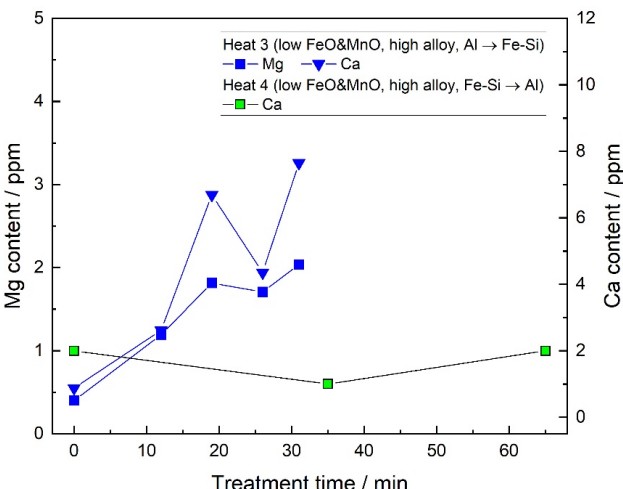

**Figure 9.** Comparison on the Ca and Mg contents between heat 3 and heat 4.

*3.2. Slag Composition*

According to the results of the chemical analysis, the contents of $SiO_2$ and MgO of the slag for all heats did not obviously change during the LF treatment. In addition, the

slag for all heats was saturated with both CaO and MgO throughout the treatment, as confirmed by comparing to the CaO–Al$_2$O$_3$–MgO phase diagram [29]. Therefore, only the CaO/Al$_2$O$_3$ (C/A) ratio and the FeO and MnO contents of slag are discussed here.

A comparison between heat 1 and heat 2 is shown in Figure 10. The C/A ratio in both heats showed similar values and increased to about 1.5 before the 1st alloying and then kept constant until the LF end. Such an increase in the C/A ratio was probably because the slag was less homogeneous after adding lime during tapping, and the sampled location of the top slag was less appropriate. Probably, for the same reason, the FeO and MnO contents showed an increase before the 1st alloying. For the FeO and MnO contents, they decreased a lot for heat 1 and little for heat 2 after the 1st alloying, due to the reduction by Al.

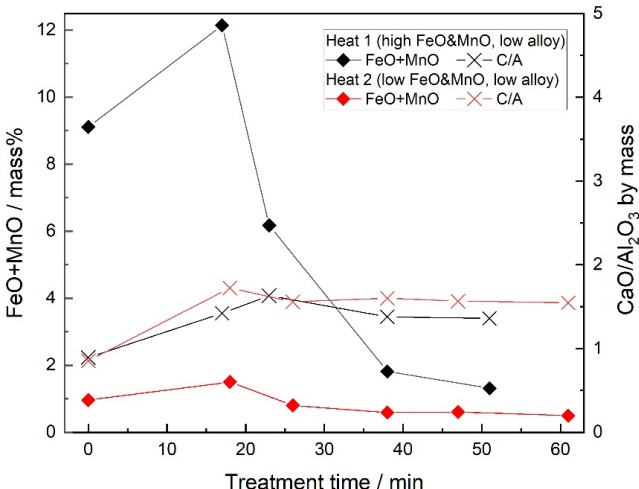

**Figure 10.** CaO/Al$_2$O$_3$ ratio and FeO and MnO contents of theslag used for heat 1 and heat 2.

The same comparison was made for heat 3 and heat 4, as shown in Figure 11. Like heat 2, the FeO and MnO contents for heat 3 and heat 4 were low and decreased a little with time. Compared to heat 1 and heat 2, the stable C/A ratio for heat 3 and 4 was higher, which was about 2. A. Harada et al. reported that the formation of the MgO–Al$_2$O$_3$ spinel inclusion was little affected by the C/A ratio when the CaO/SiO$_2$ (C/S) ratio of the slag was high [23]. In this study, the C/S ratio was always higher than five, and all slags were saturated with CaO. Therefore, the difference in the C/A ratio was neither the reason for the formation of the MgO–Al$_2$O$_3$ spinel nor the CaO–Al$_2$O$_3$ inclusion.

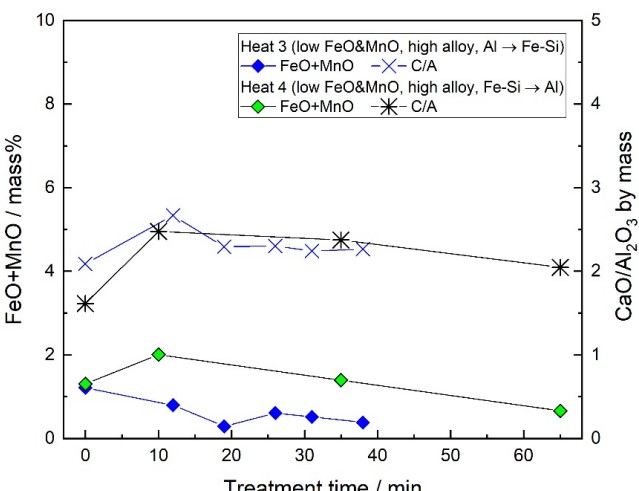

**Figure 11.** CaO/Al$_2$O$_3$ ratio and FeO and MnO contents of the slag used for heat 3 and heat 4.

### 3.3. Inclusion Composition

To clearly monitor the transformation of the inclusions, all the observed inclusions for each heat were classified as the following types. Here, $MO_{inclusion}$ indicates the $MO$ content of the inclusion in mass%, C stands for CaO, and A stands for $Al_2O_3$:

$$CA_2\text{-}C_{12}A_7; CaO_{inclusion} \geq 20\%$$

$$MgO\text{-}Al_2O_3 \text{ (MA) spinel: } MgO_{inclusion} \geq 10\% \text{ and } CaO_{inclusion} < 20\%$$

$$CA_6\text{-}CA_2: 20\% > CaO_{inclusion} > 8\% \text{ and } MgO_{inclusion} < 10\%$$

$$A\text{-}CA_6: 8\% > CaO_{inclusion} > 2\% \text{ and } MgO_{inclusion} < 10\%$$

$$Al_2O_3: CaO_{inclusion} \leq 2\% \text{ and } MgO_{inclusion} < 10\%$$

The composition of each type of inclusion, together with the average inclusion composition, were projected on the phase diagram of the CaO–MgO–Al$_2$O$_3$ system. The results of heat 1 are shown in Figure 12. Since both the Mg and Ca contents were low in molten steel, almost all the inclusions remained as Al$_2$O$_3$, even after the LF treatment, and the CaO–Al$_2$O$_3$ inclusions were scarcely observed.

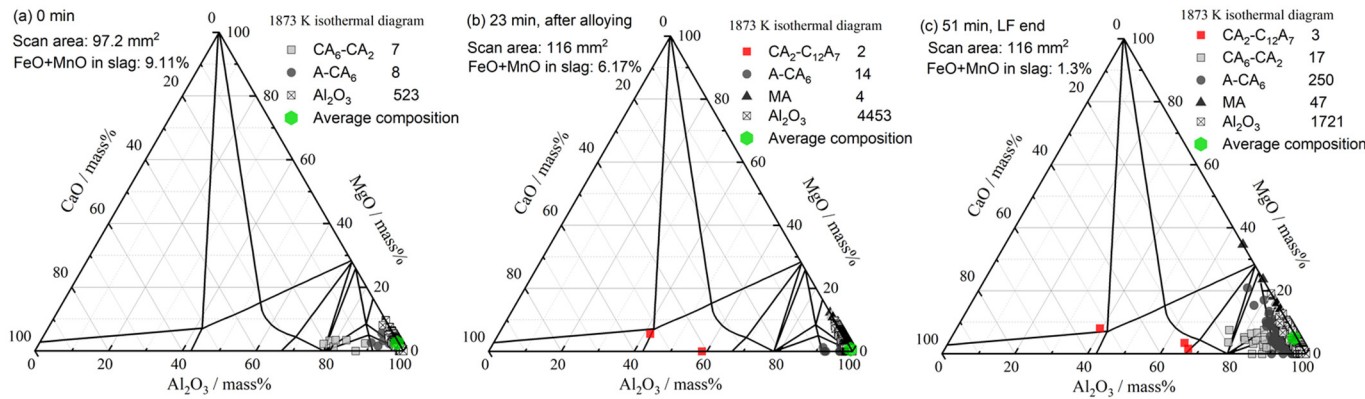

**Figure 12.** Observed inclusions in heat 1, (**a**) 0 min; (**b**) 23 min, after alloying; (**c**) 51 min, the LF end.

The inclusion compositions of heat 2 are shown in Figure 13. Different to heat 1, the initial Al$_2$O$_3$ inclusions gradually transformed into MA spinel inclusion after the 1st alloying, and almost all the inclusions became a MA spinel after the 2nd alloying and remained until the LF end. The MA spinel inclusion could form in heat 2 because of a higher Mg content of molten steel than heat 1. On the other hand, only a few CaO–Al$_2$O$_3$ inclusions were observed.

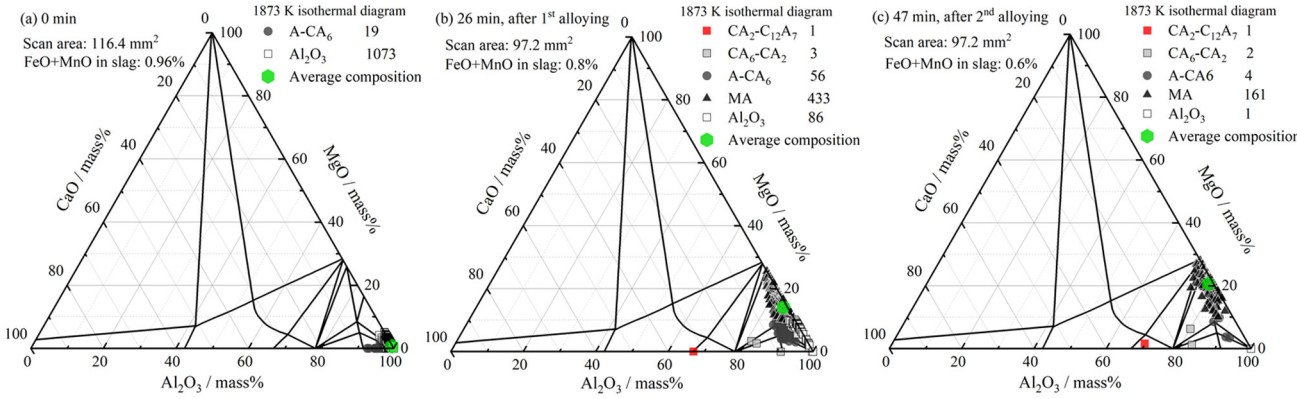

**Figure 13.** Observed inclusions in heat 2 at (**a**) 0 min, (**b**) 26 min after the 1st alloying, and (**c**) 51 min after the 2nd alloying.

Using a laboratory experiment, C. Liu et al. found that the CaO–Al$_2$O$_3$ inclusions were unlikely to form in Al–killed steel by the reaction with CaO$_{sat.}$–MgO $_{sat.}$–Al$_2$O$_3$ slag [21]. In heat 2 of the current study, the slag barely contained FeO/MnO and were also saturated with CaO and MgO, and the CaO–Al$_2$O$_3$ inclusions were barely found. Therefore, the industrial test of heat 2 confirmed the results of the laboratory study. However, it should be noticed that the Ca content of molten steel by the laboratory experiment was about 0.9 ppm when CaO–Al$_2$O$_3$ inclusions formed, and the Ca content of heat 2 was around 2 ppm, but inclusion remained as the MA spinel. Thus, the Ca content of heat 2 was probably overestimated. In heat 2, since the Ca content at the LF start was already about 1.6 ppm, the molten steel was probably contaminated by some unknown impurities during BOF tapping. The details of these contaminations will be investigated in a future study.

The inclusion composition of heat 3 is shown in Figure 14. At the start of the LF treatment, Al$_2$O$_3$, the MA spinel, and the CaO–Al$_2$O$_3$ inclusions, which contained high contents of Al$_2$O$_3$, were observed. Thus, the initial inclusions of heat 3 were different from heat 1 and heat 2, because larger amounts of the alloying materials were added during BOF tapping. However, the average inclusion composition was still Al$_2$O$_3$.

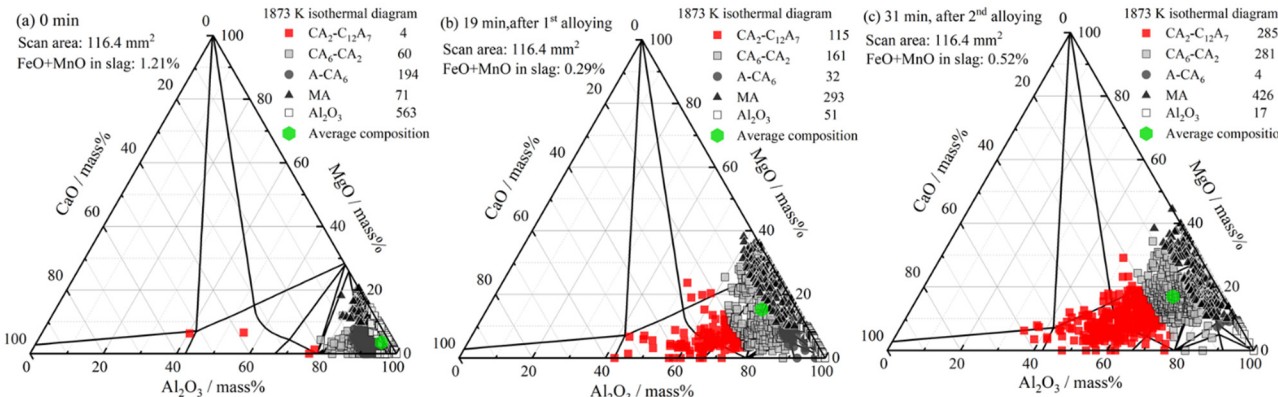

**Figure 14.** Observed inclusions in heat 3 at (**a**) 0 min, (**b**) 19 min after the 1st alloying, and (**c**) 31 min after the 2nd alloying.

After the 1st and 2nd alloying, large amounts of MA spinel inclusions and CaO–Al$_2$O$_3$ inclusions with high CaO contents were observed, and the average CaO content of the inclusions exceeded 15 mass%. The change in the inclusion composition was because both the Ca and Mg contents in molten steel were increased a lot by the addition of Fe–Si, as shown in Figure 9. Especially for the formation of CaO–Al$_2$O$_3$ inclusions, as heat 2 confirmed that it would not form by the reduction of CaO from slag, the generation of CaO–Al$_2$O$_3$ inclusions in heat 3 was only because of the Fe–Si addition.

The inclusion composition of heat 4 is shown in Figure 15. At the start of the LF treatment, both Al$_2$O$_3$ and MA spinel inclusions were observed. After the 1st alloying, which contained Fe–Si, some CaO–Al$_2$O$_3$ inclusions were observed, and the major inclusion was the MA spinel. After the 2nd alloying, almost all inclusions changed into the MA spinel. Compared to heat 3, the total oxygen content of heat 4 was close to heat 3 at the LF start, and less inclusions were counted in heat 4. Therefore, the content of dissolved oxygen in heat 4 was higher than heat 3 at the LF start. When the Fe–Si alloy was added during the 1st alloying for heat 4, the Ca content of the molten steel changed little, and large amounts of CaO–Al$_2$O$_3$ inclusions were not observed. In addition, the addition of Fe–Si for heat 4 was about half that of heat 3. Therefore, a lesser addition of Ca–containing Fe–Si alloy to a molten steel with a high content of dissolved oxygen is unlikely to cause the generation of CaO–Al$_2$O$_3$ inclusions.

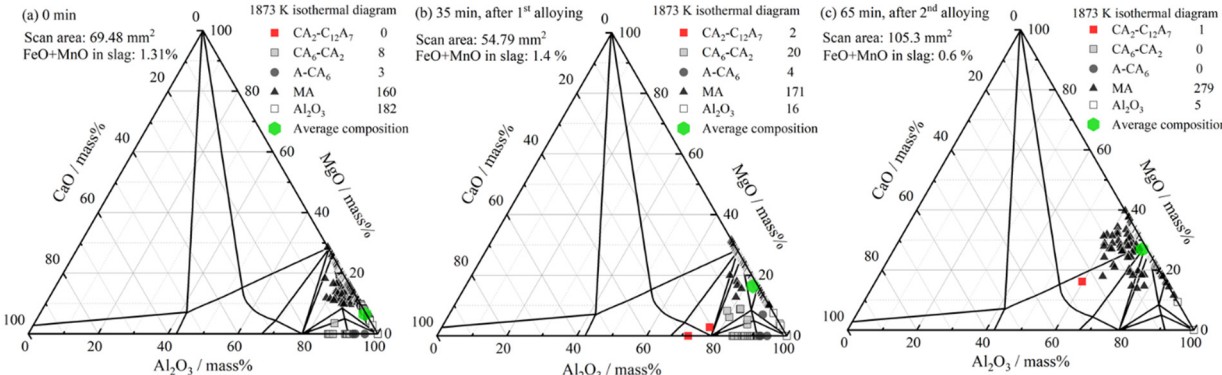

**Figure 15.** Observed inclusions in heat 4 at (**a**) 0 min, (**b**) 23 min after the 1st alloying, and (**c**) 51 min after the 2nd alloying.

## 4. Discussion

### 4.1. Formation of MgO–Al₂O₃ Spinel Inclusions

In heat 1, the FeO and MnO contents of slag at the start of the treatment were high, and the MA spinel inclusions were not observed; even the FeO and MnO contents of the top slag decreased to lower than 1 mass%. In other heats, the FeO and MnO contents of the top slag were low (<1 mass%) throughout the treatment, and large amounts of MA spinel inclusions were found. Therefore, the FeO and MnO contents of the slag affected the formation of MA spinel inclusion. The relationship between the FeO and MnO contents of the slag and the MgO content in the observed inclusions is summarized in Figure 16.

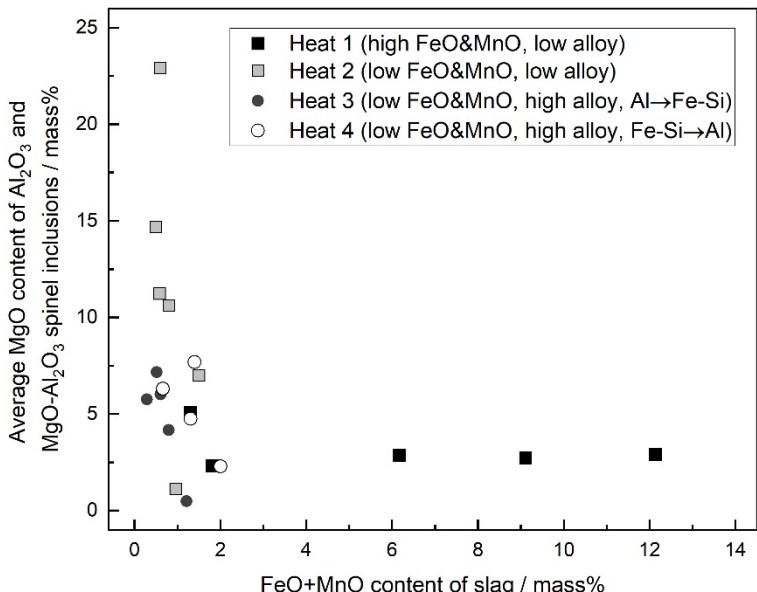

**Figure 16.** Effects of the FeO and MnO contents of the top slag on the MgO content of the observed inclusions.

The reduction of FeO and MnO in the slag by dissolved Al in molten steel occurred preferentially compared to the MgO in slag. Based on the current results, the MgO content of the inclusions began to increase when the content of FeO and MnO in the slag was below 1~2 mass%. This agreed well with other researchers [30–33]. The reason for this critical FeO and MnO content was probably due to the oxygen potential of the molten steel, which was controlled by Equation (1) [34] and Equation (2) [35] under the current experimental conditions. Here, for an easy understanding, the FeO and MnO of the slag was considered as FeO.

$$(FeO)_{slag} = Fe(l) + [O], \ logK = -6150/T + 2.604 \tag{1}$$

$$(\text{Al}_2\text{O}_3)_{\text{inclusion}} = 2[\text{Al}] + 3[\text{O}], \; logK = -45300/T + 11.62 \tag{2}$$

By combining these two equations, the activity of $\text{Al}_2\text{O}_3$ in the inclusions can be calculated for a given FeO activity of slag and an activity of dissolved Al in the molten steel. The calculated results are shown in Figure 17. In the calculations, the upper limit for the activity of $\text{Al}_2\text{O}_3$ was considered as 1. The results showed that the activity of $\text{Al}_2\text{O}_3$ started to decrease when the FeO activity was below certain values, and a high activity of dissolved Al decreased the critical activity of FeO. A decrease in the $\text{Al}_2\text{O}_3$ activity of the inclusions meant that the $\text{Al}_2\text{O}_3$ inclusions became unstable and transformed into other inclusions, like the MA spinel. The experimental results are also plotted in Figure 17. The activity of $\text{Al}_2\text{O}_3$ in the inclusion was determined using the average $\text{Al}_2\text{O}_3$ content of the observed inclusions in each heat by FactSage (); the FeO activity of the slag was also calculated by FactSage using the analyzed slag composition. By the comparison between the experimental results and thermodynamic estimation, the $\text{Al}_2\text{O}_3$ activity started to decrease when the activity of dissolved Al was between 0.002 and 0.01 and the FeO activity between 0.007 and 0.02. Based on the above discussion, it could be deduced that the FeO activity of the slag and the activity of dissolved Al of the molten steel determined the critical FeO and MnO contents shown in Figure 16. Therefore, in the practical refining process, a slight increase in the FeO and MnO activity of slag may suppress the transformation of $\text{Al}_2\text{O}_3$ inclusions into the MA spinel. However, under such operations, the total oxygen content may not be low enough for some steel that requires high cleanness.

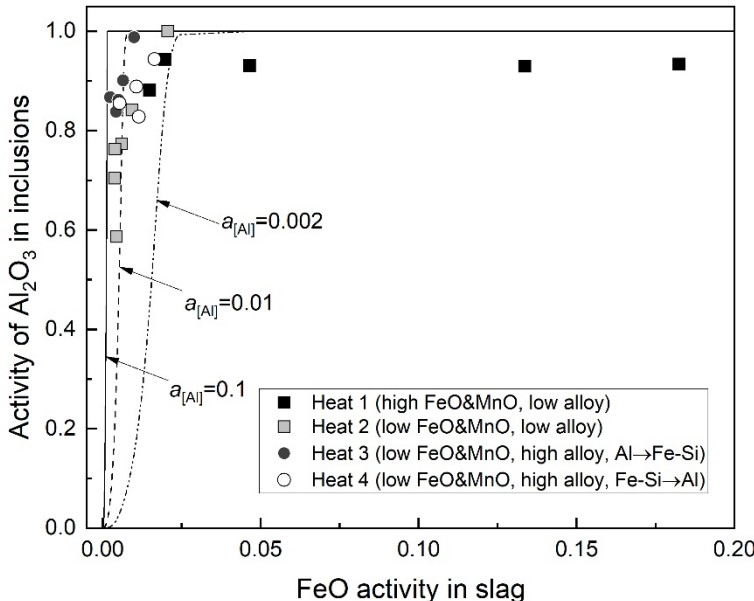

**Figure 17.** Relation between the FeO activity in the top slag and the $\text{Al}_2\text{O}_3$ activity in the inclusions.

To understand the relation between the steel composition and inclusion transformation, the results of all heats are compared to the stability diagram of $\text{MgO}/\text{MgO}\cdot\text{Al}_2\text{O}_3/\text{Al}_2\text{O}_3$, as shown in Figure 18. The details of calculating the stability diagram were introduced in a previous study [2]. In the calculations, the equilibrium constant for Al–O reported by H. Itoh et al. was used [35], and the equilibrium constant for Mg–O determined by H. Itoh et al. was used [36]. The steel compositions of heat 2 and heat 3 were located in the stable region of the spinel, and this result agrees with the observed inclusion compositions. In the case of heat 1, the observed inclusion was $\text{Al}_2\text{O}_3$, while the steel composition was located near the boundary between $\text{Al}_2\text{O}_3$ and $\text{MgO}\cdot\text{Al}_2\text{O}_3$. This disagreement was caused by the accuracy of the Mg content at an extremely low level (<0.3 ppm), and thus, the precise measurement of the Mg content, which equilibrated with $\text{MgO}\cdot\text{Al}_2\text{O}_3$ and $\text{Al}_2\text{O}_3$ in the low–Al content region, was necessary.

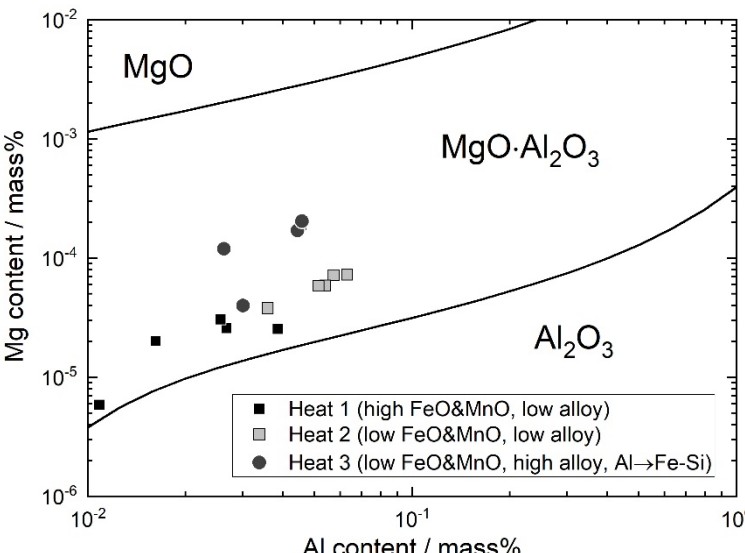

**Figure 18.** Projection of the steel compositions on the MgO/MgO·Al$_2$O$_3$/Al$_2$O$_3$ stability diagram.

### 4.2. Formation of CaO–Al$_2$O$_3$ Inclusions

The steel compositions of all heats are compared to the Al$_2$O$_3$/CaO–Al$_2$O$_3$ stability diagram, as shown in Figure 19. The calculating details of the stability diagram were introduced in a previous study. In the calculations, the equilibrium constant for Al–O reported by H. Itoh et al. was used [35], and the equilibrium constants for Ca–O determined by H. Itoh et al. [37] and by JSPS [34] were both used.

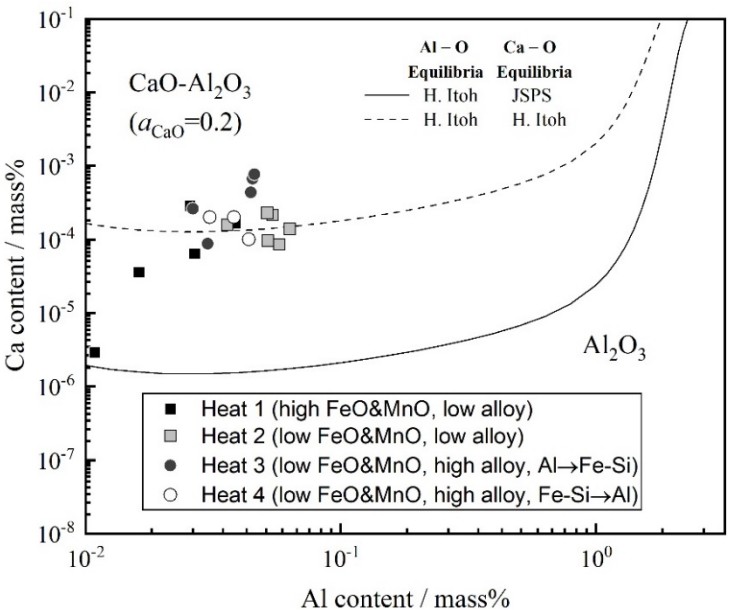

**Figure 19.** Projection of the steel compositions on the Al$_2$O$_3$/CaO–Al$_2$O$_3$ stability diagram.

Compared to the relations calculated by H. Itoh's Ca–O equilibrium constant, only the steel composition of heat 3 entered the CaO–Al$_2$O$_3$ stable region, agreeing with the observed results of the inclusions. Compared to the relations calculated by JSPS's Ca–O equilibrium constant, CaO–Al$_2$O$_3$ inclusions should formed in all heats, apparently different from the results of the inclusions. The reason for such a difference was probably because the steel compositions used to determine the Ca–O equilibrium by H. Itoh et al. were close to the current industrial tests. In heats 1 and 2, the only Ca source was the top slag; however, the reduced Ca content was never enough to cross the boundary between

$Al_2O_3$ and $CaO–Al_2O_3$, and thus, $CaO–Al_2O_3$ inclusions were not observed. In heat 4, because the dissolved oxygen at the LF start for was high and there was a small addition of Ca–containing Fe–Si, the remaining Ca content after being consuming by the dissolved oxygen in molten steel was not enough to transform the inclusions into $CaO–Al_2O_3$. Only in heat 3, when the dissolved oxygen was low and a large amount of Ca–containing Fe–Si was added, the Ca content of molten steel crossed the $Al_2O_3/CaO–Al_2O_3$ boundary, and $CaO–Al_2O_3$ inclusions formed. Based on the above discussion, we could deduce that, to decrease the formation of $CaO–Al_2O_3$ inclusions in the practical refining process, attention should be paid to the cleanness of the alloying materials or the entrapment of slag droplets if the flow at the slag/steel interface is not well–controlled but not the composition of the refining slag.

## 5. Summary

In this study, the effects of the FeO and MnO contents of a MgO and CaO–sat. slag, and the impurities of alloy materials on the transforming behavior of the inclusions, were studied through an industrial LF treatment. Some conclusions were obtained, as follows.

1. When the FeO and MnO content in the slag was high (about 10 mass%) at the start of the LF treatment, the inclusions were maintained as $Al_2O_3$ throughout the LF treatment. When the FeO and MnO content in the slag was low (less than 1~2 mass%), the $Al_2O_3$ inclusions changed to a MA spinel inclusion.

2. By the slag/steel reaction, the $CaO–Al_2O_3$ inclusions did not form under the current industrial conditions. The only Ca source for the formation of $CaO–Al_2O_3$ inclusions was confirmed to be the Ca–containing Fe–Si alloy under the current conditions. When the dissolved oxygen of the molten steel was low and a large amount of Fe–Si was added, the $Al_2O_3$ inclusions transformed into $CaO–Al_2O_3$ inclusions. However, when the dissolved oxygen was high at the LF start and less Fe–Si was added, the Ca of the Fe–Si alloy could not change the compositions of the inclusions.

**Author Contributions:** Conceptualization, C.L., F.H., S.-y.K.; methodology, C.L., F.H., X.G., S.-y.K.; formal analysis, C.L., Y.J., L.H., S.H., H.Y.; validation, X.G., F.H., S.U., S.-y.K.; writing—original draft preparation, C.L., X.G.; writing—review and editing, X.G., S.U., S.-y.K.; visualization, C.L., X.G.; supervision, F.H., S.-y.K.; All authors have read and agreed to the published version of the manuscript.

**Funding:** This research received no external funding.

**Conflicts of Interest:** The authors declare no conflict of interest.

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
