# Peer review of "Effects of Slag Composition and Impurities of Alloys on the Inclusion Transformation during Industrial Ladle Furnace Refining"

_metals, doi:10.3390/met11050763_

Round 1

Reviewer 1 Report

In the paper are presented o serie de informații privind effects of slag composition and impurities of alloys on the inclusion transformation during industrial ladle furnace refining

From the analysis of the information presented in the article, I found the following:

- The paper presents a series of results that are of interest to the scientific community:

- The introductory section needs to be improved, in order to avoid citing bibliographic sources of form [1-7]. Thus, the number of cited sources at the same time must be reduced. Attention must also be paid to the citation of bibliographic sources, so as to respect the conditions of the journal;

- The research methodology should be presented more clearly, in the sense that it is necessary to better highlight the objectives of the research and the reasons why the number of specified experiments has been established;

- The figures in which the metallographic structures are presented do not have an appropriate resolution, which means that some conclusions are not supported;

- It is necessary to specify the way in which the temperature stabilization was achieved in the range 1848 K to 1873 K;

- It is necessary a presentation of some macroscopic images for the steel samples, respectively the analyzed slags;

- The discussion part needs to be improved in order to better highlight the novelty brought by the research presented in the paper compared to other research in the field;

- In the final part of the conclusions the future research directions must be presented. The practical applications of the research could, also, be presented in conclusions.

Author Response

- The introductory section needs to be improved, in order to avoid citing bibliographic sources of form [1-7]. Thus, the number of cited sources at the same time must be reduced. Attention must also be paid to the citation of bibliographic sources, so as to respect the conditions of the journal;

Response: the introduction part has been revised again, and some new references have been added to the newly updated version.

- The research methodology should be presented more clearly, in the sense that it is necessary to better highlight the objectives of the research and the reasons why the number of specified experiments has been established;

Response: thanks for the kind suggestion. Could you be more specific about which information about the methodology should we add?

- The figures in which the metallographic structures are presented do not have an appropriate resolution, which means that some conclusions are not supported;

Response: the resolutions of all figures have been increased.

- It is necessary to specify the way in which the temperature stabilization was achieved in the range 1848 K to 1873 K;

Response: as industrial experiments, the temperature is difficult to be stabilized at a specific value. Besides, the temperature of each heat is slightly different, thus a temperature range is the best we can provide.

- It is necessary a presentation of some macroscopic images for the steel samples, respectively the analyzed slags;

Response: the photos of neither the steel nor the slag samples were taken, and unfortunately, we cannot present them. Besides, the photos of steel and slag sample has no relationship with the experimental results.

- The discussion part needs to be improved in order to better highlight the novelty brought by the research presented in the paper compared to other research in the field;

Response: thanks for the kind suggestion, but could you be more specific about what content should we add to the discussion?

- In the final part of the conclusions the future research directions must be presented. The practical applications of the research could, also, be presented in conclusions.

Response: Thanks for the kind suggestion. As we used to show the objective descriptions of experimental results as conclusions, we prefer not to add our opinion to the conclusions. However, some suggestions for the future research directions and the industrial production were added to the discussion part.

Reviewer 2 Report

Dear Authors, 

I have some comments on your work:

1) In the Introduction some references are cited in bulks. Please cite only the most important papers indicating their novelty and strong or weak points. 

2) The experiments are performed on an industrial scale. This is very important and interesting, but it is more difficult to repeat them under the same exact conditions. Even the location of samples has an influence on the results. The statistical significance in the obtained results was not taken into account. How would you evaluate the repeatability of results when the tests are conducted again?

3) What was the influence of industrial conditions on the final results? Can you compare these results with other results?

4) The novelty of the work should be indicated in the introduction and conclusions. The authors mentioned that similar experiments were conducted. So, what is the novelty? It should be also emphasized in the conclusions. 

5) What are the possible further steps in the experiments? What are the recommendations for the industry?

6) Equation 1 and 2: what is the expected error for the calculations? Are the equations suitable for all conditions or for specific conditions? 

7) what are the errors of obtained results?

Author Response

  • In the Introduction some references are cited in bulks. Please cite only the most important papers indicating their novelty and strong or weak points. 

Response: the introduction part has been revised again, and some new references have been added to the newly updated version.

  • The experiments are performed on an industrial scale. This is very important and interesting, but it is more difficult to repeat them under the same exact conditions. Even the location of samples has an influence on the results. The statistical significance in the obtained results was not taken into account. How would you evaluate the repeatability of results when the tests are conducted again?

Response: Thank you very much for the comments. The repeatability of experimental results is an important issue. Thus, we used a well-tested sampling method and procedure as introduced by [ref. 28] to ensure the sample is representative. Then, the analysis methods have also been well established and been used many times, the accuracy of the data is acceptable.

  • What was the influence of industrial conditions on the final results? Can you compare these results with other results?

Response: as the experimental conditions for industrial and laboratory-scale are completely different, we prefer not to compare them, though we have done many laboratory-scale experiments and confirmed similar phenomenon (such as the reduction of CaO from slag is hardly to generate CaO-Al2O3-type inclusion). This is one of the important meaning that we conduct these industrial experiments, because the laboratory-scale is not representative for industrial conditions, and we need to confirm similar phenomenon also happens in practical production.

  • The novelty of the work should be indicated in the introduction and conclusions. The authors mentioned that similar experiments were conducted. So, what is the novelty? It should be also emphasized in the conclusions. 

Response: as mentioned in last response, as the laboratory-scale experiments are very different from industrial production, it is necessary to confirm our previous conclusions using industrial test. Such information has been provided in the introduction part.

  • What are the possible further steps in the experiments? What are the recommendations for the industry?

Response: some suggestions for the future research directions and the industrial production were added to the discussion part.

  • Equation 1 and 2: what is the expected error for the calculations? Are the equations suitable for all conditions or for specific conditions? 

Response: The values in Eq. (1) and (2) are not calculated by us, but reported equilibrium reaction constants. In addition, these two reactions are widely used in many papers, but unfortunately there is no information about the calculation errors.

  • what are the errors of obtained results?

Response: Since the sampling method as well as the analysis method have been used, the errors are considered small, and the differences among results are large enough for discussion.

Reviewer 3 Report

Review of “Effects of slag composition and impurities of alloys on the inclusion transformation during Industrial ladle furnace refining”

This industrial work seems very interesting for research community, overall a nice work but in my opinion the results achieved should be discussed in more details against the literature data. Other comments are listed below :

It is required to define all the acronym’s before their first appearance in text ; i.e.. “LF”- line 25

Very few references are from new literature, therefore I suggest to improve the stare of art with more much newer references !

The English should eb considerable improved; just an example “In the refining process of clean steel,- type inclusions sometimes formed after Al deoxidation”

What does means the arow and number in Figure 4 and 5 ?

Author Response

It is required to define all the acronym’s before their first appearance in text ; i.e.. “LF”- line 25

Response: Thanks for the kind suggestion. All abbreviations at their first appearance have been explained.

Very few references are from new literature, therefore I suggest to improve the stare of art with more much newer references !

Response: some new references have been added to the list.

The English should eb considerable improved; just an example “In the refining process of clean steel,- type inclusions sometimes formed after Al deoxidation”

Response: thanks for the kind suggestion, the expressions have been carefully checked again. In addition, about the expression “-type inclusion”, we have used this expression in previous papers (ISIJ Int., 2020, 60, 1835-1848), and this expression have been confirmed by a professional agency of English proofreading.

What does the arrow and number in Figure 4 and 5 mean?

Response: an arrow is pointing to a location/phase which has been analyzed, and the number corresponds to the composition shown in the tables.

Round 2

Reviewer 1 Report

The authors revised their manuscript according to my suggestions. Thus the manuscript can be accepted for publication.

Reviewer 2 Report

Dear Authors, 

I accept your answers. 

Regards, 

Reviewer 

Reviewer 3 Report

.